# Difference in Methylation and Expression of Brain-Derived Neurotrophic Factor in Alzheimer’s Disease and Mild Cognitive Impairment

**DOI:** 10.3390/biomedicines11020235

**Published:** 2023-01-17

**Authors:** Katarina Kouter, Matea Nikolac Perkovic, Gordana Nedic Erjavec, Tina Milos, Lucija Tudor, Suzana Uzun, Ninoslav Mimica, Nela Pivac, Alja Videtic Paska

**Affiliations:** 1Institute of Biochemistry and Molecular Genetics, Faculty of Medicine, University of Ljubljana, SI-1000 Ljubljana, Slovenia; 2Laboratory for Molecular Neuropsychiatry, Division of Molecular Medicine, Ruder Boskovic Institute, 10000 Zagreb, Croatia; 3Department for Biological Psychiatry and Psychogeriatrics, University Psychiatric Hospital Vrapce, 10090 Zagreb, Croatia; 4School of Medicine, University of Zagreb, 10000 Zagreb, Croatia

**Keywords:** epigenetics, DNA methylation, gene expression, dementia, Alzheimer’s dementia, brain-derived neurotrophic factor, BDNF, catechol-o-methyltransferase, COMT, cognitive impairment

## Abstract

Due to the increasing number of progressive dementias in the population, numerous studies are being conducted that seek to determine risk factors, biomarkers and pathological mechanisms that could help to differentiate between normal symptoms of aging, mild cognitive impairment (MCI) and dementia. The aim of this study was to investigate the possible association of levels of *BDNF* and *COMT* gene expression and methylation in peripheral blood cells with the development of Alzheimer’s disease (AD). Our results revealed higher expression levels of *BDNF* (*p* < 0.001) in MCI subjects compared to individuals diagnosed with AD. However, no difference in *COMT* gene expression (*p* = 0.366) was detected. DNA methylation of the CpG islands and other sequences with potential effects on gene expression regulation revealed just one region (BDNF_9) in the *BDNF* gene (*p* = 0.078) with marginally lower levels of methylation in the AD compared to MCI subjects. Here, we show that the level of *BDNF* expression in the periphery is decreased in subjects with AD compared to individuals with MCI. The combined results from the gene expression analysis and DNA methylation analysis point to the potential of BDNF as a marker that could help distinguish between MCI and AD patients.

## 1. Introduction

Alzheimer’s disease (AD) is a progressive neurodegenerative disease characterized by memory loss and cognitive dysfunction. AD is the most common type of dementia and there are around five million new cases of AD every year [1]. Since the occurrence of AD is strongly associated with age, the disease burden is expected to increase with the aging of populations and increased life expectancy [2]. There are two subtypes of AD. The hereditary form or early onset AD is characterized by onset of the disease before the age of 65, while the late onset or sporadic form of the disease occurs after the age of 65 and is the most prevalent type of AD [3,4]. The prodromal phase of neurodegenerative diseases consists of the insufficient recovery of neurons, coupled with an increase in inflammatory cytokines and neuronal damage, leading to decreased resistance of neuroplasticity in neurodegenerative diseases [5,6]. During the course of AD, different brain regions are affected. AD-associated changes usually form first in the hippocampus and later spread to other brain regions [7,8]. Associated with the hippocampus and also implicated in AD neuropathology are the prefrontal cortex and amygdala; these are brain regions highly involved in learning, memory formation, cognition and fear processing and conditioning [9]. Similar brain regions are involved in depressive states, which are often closely related to AD, although the question remains unanswered as to whether depression is merely a risk factor for AD or part of the prodromal stage. Both depression and AD have in common the increase in inflammatory cytokines, which could explain the disturbances in the neurotransmitter systems in depression, whereas in AD, inflammatory cytokines could affect the function of microglial cells and amyloid peptide metabolism enzymes [6].

The characteristic neuropathological hallmarks of AD are the accumulation of beta-amyloid and the generation of senile plaques, along with the formation of neurofibrillary tangles due to hyperphosphorylation of the tau protein, resulting in the loss of synapses [10,11]. These changes are accompanied by neuronal death and damage to brain tissue. Other changes include dysregulated serotonin and dopamine neurotransmission, imbalance of the enzymes involved in the generation and removal of reactive oxygen species, a reduction in cholinergic function and changes in neurotrophic factors, such as brain-derived neurotrophic factor (BDNF) [12]. In recent years, mitochondria have gained importance in the etiology of various biological conditions. While mitochondrial alterations are part of the normal aging process, there is increasing evidence that mitochondria are involved in neurodegeneration as they are crucial to the normal neurotransmission activity of neurons [13]. Mitochondria are an important source of reactive oxygen species that, in conjunction with impaired antioxidant defense systems, can lead to the formation of oxidative stress. Both mitochondrial structure and function appear to be affected in AD, possibly due to oxidative stress. Free radicals could affect the activity of enzymes involved in amyloid β formation [14]. Additional mitochondrial dysfunction has been shown to be associated with AD, with studies examining genetic variations and epigenetic mechanisms in mitochondrial DNA [15], as well as its involvement in metabolic pathways such as tryptophan–kynurenine metabolism [13]. AD is a multifactorial disease where genetic, environmental and behavioral factors play an important role, and some of the risk factors include aging, brain trauma, neuroinflammation and oxidative stress [16,17]. There are numerous genes that have been investigated for their association with AD, but the research results for the majority of these genes are inconsistent. The only gene that is currently being used for predicting the possibility of the development of AD is apolipoprotein E (ApoE). Other gene candidates possibly associated with the development of AD are the genes coding for the amyloid precursor protein, presenilin-1 and presenilin-2; genes involved in neurotransmission systems such as genes coding for BDNF; catechol-o-methyltransferase (COMT); serotonin transporter; and dopaminergic receptors [18,19]. 

BDNF belongs to the family of proteins that promote neuronal survival, development and function. It is involved in neurogenesis, neurotransmission, proliferation, regeneration, the promotion of synaptic growth and the modulation of synaptic plasticity [20]. BDNF plays an important role in modulating cognition, learning and memory. It is highly expressed and distributed throughout the central nervous system (CNS), especially in the hippocampal area and cerebral cortex [21], and it is essential in the survival and function of hippocampal, cortical, cholinergic and dopaminergic neurons [22,23,24]. Therefore, it has been suggested that BDNF could be associated with the etiopathogenesis of various brain disorders including schizophrenia, depression, addiction, eating disorders, post-traumatic stress disorder and neurodegenerative disorders, such as AD [20,25,26]. Decreased mRNA levels of *BDNF* were reported in postmortem brain samples, as were changes in BDNF/TrkB signaling, which indicates the impairment of BDNF and its receptor, TrkB, in AD [27]. Additionally, decreased protein levels of BDNF were found in the hippocampal and cortical areas of AD brains [28,29]. In patients with AD, significantly lower serum and cerebrospinal fluid concentrations of BDNF were reported [30,31]. A reduction in BDNF was associated with neuroinflammation, the accumulation of beta-amyloid protein, tau phosphorylation and neuronal apoptosis [32]. Interestingly, it has been shown that BDNF overexpression alleviates neuronal loss, synaptic degradation and behavioral deficits [33]. In addition, higher levels of BDNF have been correlated with better cognitive function in AD [34]. These findings suggest that BDNF has a role in the etiology of AD, and thus, has the potential to be considered as a biomarker for the early detection of AD. 

COMT acts as a key mediator in various brain functions and has a prominent role in catecholamine metabolism [35], with a central role in dopamine degradation in the prefrontal cortex [36]. COMT regulates dopamine prefrontal levels and its altered activity has been associated with different brain disorders and cognitive dysfunction [37]. It is widely expressed in the CNS and in the periphery (lungs, liver, kidney and intestines). Genetic variants of COMT influence cognitive functions, in particular, working memory, learning processes, emotional regulation [38,39], cognitive control and intelligence [40]. There are several functional polymorphisms of the *COMT* gene, but the most commonly investigated is a G/A substitution, which results in a change in amino acid from Valine (Val or G) to Methionine (Met or A) (Val158/108Met (rs4680) polymorphism) [41]. The Met allele is associated with low enzymatic activity, whereas the Val allele is associated with higher activity. Because of its lower activity, studies report that the Met variant is associated with increased dopaminergic stimulation of postsynaptic neurons in the prefrontal cortex [42]. Therefore, Val158/108Met polymorphism affects enzyme activity and modulates dopamine signaling in the prefrontal cortex and it is associated with impairments in cognitive processes. Different studies have reported mixed findings regarding the relationship between Val158/108Met polymorphism and the risk of developing AD. Therefore, a better understanding of the role of COMT in cognitive processes could improve the development of potential therapeutic strategies for patients with AD and other brain disorders. 

The aim of this study was to determine the potential epigenetic modifications and mRNA expression levels of *BDNF* and *COMT* at the periphery and to determine combinations of biomarkers that would yield the largest predictive values for differentiation between patients with AD and MCI. These results could lead to the introduction of potential new biomarkers in diagnostic procedures in clinical practice and might improve the treatment of patients with AD and MCI.

## 2. Materials and Methods

### 2.1. Participants

A total of 248 subjects (90 males) were included in the study and divided, according to diagnosis, into subjects with AD (n = 162) and subjects with MCI (n = 86). Out of 248 subjects, 160 of them (40.0% men) were included in the expression analysis study, 74 of which had AD and 86 of which had been diagnosed with MCI. All participants were recruited from the Psychiatric Clinic of Vrapce, Zagreb, Croatia. 

The diagnosis of AD and MCI was based on DSM-5 [43] criteria and the criteria of the National Institute of Neurological and Communication Disorders and Stroke, which is part of the American National Institute of Health (NINCDS-ADRDA; the National Institute of Neurological and Disorders and Stroke and the Alzheimer’s Disease and Related Disorders Association). The age of onset for patients with AD was 66.8 ± 9.9 years, and for MCI subjects, 63.0 ± 10.6 years. The duration of disease was 2.5 ± 1.8 years for AD and 2.1 ± 1.5 years for MCI. The cognitive abilities of all participants were evaluated using a Mini-Mental State Examination (MMSE) [44,45] and a Clock Drawing Test (CDT) [46,47].

The inclusion criteria were as follows: in- and out-patients who had signed informed written consent, and subjects who had not previously taken any antidementia drugs and were not related to each other. All subjects diagnosed with vascular or mixed dementia, tumors or inflammatory diseases of the central nervous system, brain trauma, systemic metabolic diseases or other psychiatric or neurological diseases (e.g., Huntington’s disease or frontotemporal dementia) were excluded from the mentioned research.

The study was carried out in line with the Declaration of Helsinki [48] and approved by the Ethics Committee of the Psychiatry Clinic of Vrapce, Zagreb, Croatia. The study procedures were explained in detail to the patients with AD and subjects with MCI and their caregivers. All participants signed informed consent prior to participating in the study.

### 2.2. Gene Expression

#### 2.2.1. Blood Sample Collection

Whole blood samples (8.5 mL) were collected at 8 a.m., following an overnight fast, in yellow-top BD Vacutainer™ tubes with 1.5 mL of acid citrate dextrose anticoagulant. Sampling was performed during routine laboratory visits. 

Centrifugation (3 min, 1100× *g*) was used to separate the plasma from the whole blood sample and the remaining part of the blood sample was used for the isolation of total RNA and DNA.

#### 2.2.2. RNA Isolation and Reverse Transcription

RNA extraction was performed using the PureLink RNA Mini Kit (Thermo Fisher Scientific, Life Technologies, Waltham, MA, USA) and random hexamers according to the manufacturer’s instructions. The quantity and quality of the RNA samples were determined using a NanoPhotometer^®^ C40 (Implen, München, GermanyUSA). Total RNA (500 ng/20 µL) was transcribed into complementary DNA (cDNA) using a RevertAid First Strand cDNA Synthesis Kit (Thermo Fisher Scientific, Life Technologies, Waltham, MA, USA) according to the manufacturer’s protocols. The reaction parameters were: 25 °C for 5 min, 60 °C for 60 min and 70 °C for 5 min. The cDNA samples were stored at −80 °C until real-time PCR analysis was performed.

#### 2.2.3. Real-Time PCR Analysis

The gene expression, for both *BDNF* and *COMT*, was determined using the ABI 7300 Real-Time PCR System (Applied Biosystems, Foster City, CA, USA) using TaqMan Gene Expression Assays—Hs02718934_s1 for BDNF and Hs00241349_m1 for COMT (Applied Biosystems, Foster City, CA, USA). The glyceraldehyde-3-phosphate dehydrogenase (GADPH) gene was used as a housekeeping gene to normalize the expression of targeted genes (Hs00266705_g1). The amplification reactions were performed in triplicate. After the initial denaturation at 95 °C for 10 min, 50 cycles of 95 °C for 15s and 60 °C for 1 min followed. Cycle threshold (Ct) values were determined using SDS software v1.3 (Applied Biosystems, Foster City, CA, USA).

#### 2.2.4. Statistical Analysis

The results were evaluated using Sigma Stat 3.5 (Jandel Scientific Corp., San Jose, CA, USA). A Kolmogorov–Smirnov test was used to evaluate the normality of data distribution. Due to the non-normal distribution of the analyzed variables, non-parametric tests were applied and the results were expressed as the median and range (min-max). A comparison between the groups was performed using a Mann–Whitney U test, and the correlation was evaluated using Spearman’s correlation coefficient. All tests were two-tailed, and α was set at 0.05. G∗Power 3 Software [49] was used to calculate the needed sample size and statistical power. With the expected effect size = 0.50 and statistical power set to 0.85, the required sample size was n = 154 for the Mann–Whitney U test and n=150 for the correlation. Our study included a total of 160 subjects, which is above the sample size needed to detect differences between groups.

### 2.3. DNA Methylation

#### 2.3.1. DNA Isolation and Bisulfite Conversion

The extraction of genomic DNA was carried out using the PureLink™ Genomic DNA Mini Kit (Invitrogen, Carlsbad, CA, USA) according to the manufacturer’s protocol. The DNA samples were stored at −20 °C at the Ruder Boskovic Institute (RBI) until further analysis. Bisulfite conversion was performed on 800 ng of DNA using an EpiTect Fast Bisulfite Kit (Qiagen, Venlo, The Netherlands) as per the manufacturer’s protocol.

#### 2.3.2. Primer Design 

CpG island (CGI) sequences of the *BDNF* and *COMT* genes with additional 500-base-pair flanking regions upstream and downstream of the CGI were obtained from the UCSC genome browser (Homo sapiens version hg19, http://genome.ucsc.edu/ (accessed on 26.03.2019)) [50]. For primer design, Methyl Primer Express (v1.0, https://resource.thermofisher.com/page/WE28396_2/ (accessed on 27 March 2019)) was used. Primer pairs were designed to amplify approximately 300-base-pair-long regions. For the *BDNF* gene, the CGIs in the promoter region or the first and fourth exon were interrogated. For the *COMT* gene, the GCI around promotor 2 was analyzed, as were two regions outside the GCI but with a potential impact on gene expression: the region around promotor 1 and the region around the single-nucleotide polymorphism rs4680. 

The specificity and properties of the designed primers were determined using BiSearch (v2.63, http://bisearch.enzim.hu/ (accessed on 10 May 2019)) and IDT oligo analyzer (https://eu.idtdna.com/pages/tools/oligoanalyzer (accessed on 10 May 2019)). To the DNA-sequence-specific primer, the Illumina adapter overhang sequences were added to the 5′ end. The amplicon positions and lengths and the number of CpGs are listed in Table 1.

#### 2.3.3. Amplicon Generation and Sequencing 

The amplicon library was prepared according to the Illumina 16S protocol with some modifications [51]. The amplification of target sequences was performed with two rounds of PCR. In the first round, the target sequences were generated (complete primer pair sequences in Appendix A). The PCR reactions were carried out in 25 μL reactions, composed of 12.5 μL of KAPA HiFi HotStart Uracil+ ReadyMix (Roche, KAPA Biosystems Ltd, Cape Town, South Africa), 1 μM of primers and 20 ng of DNA. The PCR protocol consisted of activation for 5 min at 95 °C and 35 cycles of amplification (denaturation for 30 s at 98 °C, annealing for 15 s at a primer pair-dependent temperature and extension for 15 s at 72 °C), followed by final extension for 1 min at 72 °C, and holding at 4 °C. The annealing temperatures are listed in Appendix A. After the first round of amplification, the fragments were visualized via 2% agarose gel electrophoresis to determine the generation of amplicons of suitable lengths. The cleanup of shorter, unspecific fragments was performed using AMPure XP beads (Beckman Coulter, Brea, CA, USA). In the next step, we combined all amplicons of each subject into an equimolar pool (concentrations were measured using Quant-iT PicoGreen dsDNA (Thermo Scientific, Life Technologies, Waltham, MA, USA).

The second round of PCR was performed on the pooled samples in order to add specific identifiers for each subject to enable multiplexing. For the second round of PCR, Nextera XT v2 index set A and set D primers (Illumina, San Diego, CA, USA) were used. PCR reactions were carried out in a total volume of 50 μL, composed of 25 μL of KAPA HiFi HotStart Uracil+ ReadyMix (Roche, Basel, Switzerland), Nextera XT v2 primers and 4 ng of the equimolar amplicon pool. The PCR protocol consisted of activation for 45 s at 98 °C, 10 cycles of amplification (denaturation for 15 s at 98 °C, annealing for 30 s at 55 °C and extension for 30 s at 72 °C), followed by final extension for 1 min at 72 °C. The PCR products were again visualized via 2% agarose gel electrophoresis to determine the generation of amplicons of suitable lengths. 

#### 2.3.4. Library Preparation and Sequencing

Following the second round of PCR amplification and size selection (AMPure XP paramagnetic beads, Beckman Coulter, Brea, CA, USA), the concentration of subject libraries was measured using an ultrasensitive fluorescent nucleic acid stain for quantitating double-stranded DNA (PicoGreen dsDNA quantitation reagent, Thermo Fisher, Waltham, MA, USA). Libraries from individual subjects were equimolarly pooled into a final library with a 10 nM molar concentration. The final library was diluted and denatured following the Illumina MiniSeq System Denature and Dilute Libraries Guide recommendations. The library was sequenced using the Illumina MiniSeq sequencer and MiniSeq Mid Output Kit (300-cycles), using 150 bp paired-end reads.

#### 2.3.5. Bioinformatic and Statistical Analysis

Raw sequencing reads in FASTQ format were assessed for quality using the FastQC tool [52]. Bases of insufficient quality (Q score below 30) and adapter sequences were trimmed using Trim galore [53]. Trimmed sequences were aligned to the reference genome UCSC genome browser (Homo sapiens version hg19) using Bismark [54]. Aligned reads were further analyzed using the R environment [55], methylKit package [56] and methylSig package [57], where data were corrected for age and multiple testing. Differentially methylated CpGs (DMC) were identified via DNA methylation percentage comparison for each CpG cytosine between our two test groups. The average amplicon DNA methylation values were calculated using the values of all CpGs residing in said amplicon, and compared between both our studied groups of subjects. The normality of CpG value distribution was assessed using the Shapiro–Wilk normality test. As the distribution was nonparametric for most of the amplicon data, the differences in the percentages of amplicon mean methylation between the groups were calculated using the Mann–Whitney U test and Benjamini–Hochberg correction. Corrected *p*-values under 0.05 were deemed statistically significant.

## 3. Results

### 3.1. Participants

A total of 248 subjects, both male and female, were included in the study and divided, according to diagnosis, into subjects with AD and subjects with MCI. The investigated groups did not differ significantly in the proportion of male and female participants (χ^2^ = 0.71; df = 1; *p* = 0.400).

The demographic and clinical data for the participants are shown in Table 2. The distribution of all demographic and clinical variables was tested using the Kolmogorov–Smirnov test. Considering deviations from the normal distribution in the case of all examined demographic and clinical parameters, the non-parametric Mann–Whitney U test was used for the comparison between subject groups (Table 2).

Subjects diagnosed with AD and individuals with MCI differed significantly in age, since subjects with AD were significantly (*p* ≤ 0.001) older than the MCI group (Table 2). The comparison of subjects with respect to MMSE and CDT score confirmed that the investigated groups differed significantly (*p* ≤ 0.001) in their cognitive abilities (Table 2). There was no significant difference in the other clinical and demographic data between subjects diagnosed with AD and subjects with MCI (Table 2).

### 3.2. BDNF Gene Expression

For the relative quantification of *BDNF* expression, the comparative Ct (ΔΔCt) method was used, that is, the relative expression of the *BDNF* gene was shown as a 2^–ΔΔCt^ value. Since the *BDNF* expression deviated from the normal distribution, non-parametric tests were used for statistical analyses.

Considering the potential influence of individual demographic and clinical parameters on *BDNF* expression, the correlation (Spearman’s correlation coefficient) was examined between all demographic and clinical characteristics and the expression of *BDNF* in both groups of subjects (Appendix A). The results (Appendix A) revealed no significant association between individual demographic and clinical parameters and the level of *BDNF* expression in either subject group. Only in the case of HDL was a significant correlation (*p* = 0.018) with the *BDNF* expression detected, but only in subjects with MCI (Appendix A). The possible influence of gender on *BDNF* expression was analyzed in both groups of subjects using the Mann–Whitney U test. In the case of subjects diagnosed with AD, there was no significant difference in *BDNF* expression between men and women (U = 582.0; *p* = 0.555), and the same trend was observed in subjects with MCI (U = 900.5; *p* = 0.958). Due to these results, there was no need for further correction of *BDNF* expression levels for the influence of age, gender, BMI, waist circumference, total cholesterol, HDL, LDL, triglycerides and fasting glucose.

The level of *BDNF* expression between subjects with MCI and subjects with AD was compared using the Mann–Whitney U test. The results showed a significant difference (U = 1964.5; *p* < 0.001) in *BDNF* expression between the two groups of subjects (Figure 1). The aforementioned difference arises from the reduced expression of *BDNF* in subjects with AD (0.42; 0.02–4.39) compared to individuals with MCI (0.84; 0.17–14.38) (Figure 1).

For the relative quantification of *BDNF* gene expression, the comparative Ct (ΔΔCt) method was used. The relative expression of the *BDNF* gene was shown as a 2^–ΔΔCt^ value.

### 3.3. COMT Gene Expression

Similarly, as in the case of *BDNF* expression, for the relative quantification of *COMT* expression, the comparative Ct (ΔΔCt) method was used, that is, the relative expression of the *COMT* gene was shown as a 2^–ΔΔCt^ value. Since *COMT* expression deviated from the normal distribution, non-parametric tests were used for statistical analyses.

In the evaluation of the potential influence of individual demographic and clinical parameters on *COMT* expression, we examined the correlation (Spearman’s correlation coefficient) between all demographic and clinical characteristics and the expression of *COMT* in subjects with AD or MCI (Appendix A). The results (Appendix A) showed no significant association between individual demographic and clinical parameters and the level of *COMT* expression in either subject group (Appendix A). The possible influence of gender on *COMT* expression was analyzed in the groups with AD and MCI using the Mann–Whitney U test. Within subjects diagnosed with AD, there was no significant difference in *COMT* expression between men and women (U = 603.0; *p* = 0.724), and no significant sex-related differences were found in subjects with MCI (U = 2522.0; *p* = 0.055). Because of these results, we did not correct *COMT* expression levels for the influence of age, gender, BMI, waist circumference, total cholesterol, HDL, LDL, triglycerides and fasting glucose.

The level of *COMT* expression between subjects with MCI and AD was compared using the Mann–Whitney U test. No significant difference (U = 2918.0; *p* = 0.366) was detected in *COMT* expression between subjects diagnosed with AD (0.94; 0.02–6.89) and the MCI group (1.02; 0.10–5.18) (Figure 2). 

For the relative quantification of *COMT* gene expression, the comparative Ct (ΔΔCt) method was used. The relative expression of the *BDNF* gene was shown as a 2^–ΔΔCt^ value.

### 3.4. DNA Methylation

We investigated the DNA methylation levels of 12 amplicons altogether; nine of them investigated the DNA methylation status of the *BDNF* gene, and three of them investigated the DNA methylation status of the *COMT* gene. As DNA methylation can be closely associated with the aging process, the results were age-corrected. Since the DNA methylation levels deviated from the normal distribution, nonparametric tests were used for statistical analyses. The results are presented in Table 3.

Upon analyzing the mean DNA methylation levels per amplicon, no statistically significant changes were observed between the average DNA methylation levels between subjects with AD and subjects with MCI. When comparing the DNA methylation status of CpGs between subjects with AD and MCI, we did detect multiple single-base-pair DMCs. We detected the highest number of DMCs in the *BDNF* gene amplicon BDNF_9, where 8 out of the 23 investigated CpG sites showed statistically significant differences in DNA methylation levels. However, when looking at the mean DNA methylation level, the mean level of BDNF_9 only showed a trend towards statistical significance. The differences in the DNA methylation levels between subjects with AD and subjects with MCI were modest (under 1%), which opens the question of their biological relevance. 

## 4. Discussion

Due to the increase in the proportion of older people in the population, there is an increase in the prevalence and incidence of progressive dementia. There are many unanswered questions related to the etiology and pathology of dementia, and therefore, numerous studies are being conducted that, through various approaches, seek to determine the risk factors, biomarkers and pathological mechanisms that could help to differentiate between normal symptoms of aging, MCI and dementia. Owing to the lack of reliable data in the literature, the aim of this paper was to investigate whether there is a connection between the levels of *BDNF* and *COMT* gene expression and methylation in peripheral blood cells with the development of AD. Our results show higher expression levels of the *BDNF* in MCI subjects compared to individuals diagnosed with AD. However, no difference in *COMT* gene expression was detected between the two groups of subjects. For DNA methylation of the CGIs and other sequences with potential effects on gene expression regulation, just one region (BDNF_9) was determined to be borderline statistically significant in the *BDNF* gene (*p*-value 0.0778, which had a lower level of methylation in the AD subjects). As the difference in the methylation level between AD and MCI was slight, whether it can impact gene expression levels is questionable. For the *COMT* gene, no statistically significant differences were observed in the methylation levels between AD and MCI subjects.

The neuroprotective effects of BDNF, its role in the regulation of neurotransmitter systems and its function in the pathogenesis of AD have been extensively studied. The *BDNF* gene is located on chromosome 11, it contains 11 exons, and is about 70 kb long [58]. The transcription of *BDNF* is controlled through nine different promoters which, combined with the process of alternative intron splicing, results in more than 30 known transcript isoforms that are expressed differently in different tissues and in response to various stimuli [58]. All the transcript isoforms of the *BDNF* gene give the same protein product, which is encoded by exon IX. BDNF is not only located in the CNS, but is also present in large quantities in platelets [59], epithelial and vascular cells, muscle cells, macrophages and leukocytes [60,61,62]. Research on BDNF as a potential peripheral biomarker of AD is supported by its ability to cross the blood–brain barrier [63,64] and by studies providing evidence of a correlation between the BDNF levels in the peripheral blood and the amount of BDNF in the CNS [65,66]. In the case of *BDNF*, our results suggest that the expression of this gene was reduced in AD patients compared to subjects diagnosed with MCI. Previous studies have shown reduced BDNF mRNA and protein expression in different brain regions of AD patients and in the substantia nigra of subjects diagnosed with Parkinson’s disease [29,67]. *BDNF* mRNA expression was found to be reduced in the hippocampus and temporal cortex of individuals with AD, suggesting that BDNF could be involved in progressive neuron atrophy accompanying dementia [68]. Reduced *BDNF* mRNA expression was also detected in the parietal and entorhinal cortex of subjects diagnosed with AD, and the observed down-regulation of *BDNF* mRNA expression was associated with increased neuritic plaque pathology [69,70]. Peng and colleagues demonstrated that a reduction in both the precursor (pro-BDNF) and the mature BDNF (mBDNF) level already occurs in the early stages of AD progression [29], even though these two forms of BDNF bind to different receptors and exert opposing biologic functions. The obtained results are in accordance with different studies that indicate a reduced level of BDNF in the brain tissue of patients with AD [71,72,73,74]. However, it should be pointed out that that there are conflicting results about BDNF protein expression, which are not consistent with the results seen at the mRNA level [75]. These discrepancies could be due to the fact that it is very difficult to differentiate pro-BDNF and mBDNF using some ELISA methods for detection and because different studies include patients in different stages of AD [76,77]. With the detection of BDNF in CSF samples, the biggest problem is its low abundance [78]. The reduced level of *BDNF* mRNA in subjects with AD may be related to the fact that this group is, on average, was significantly older than the group of subjects diagnosed with MCI. Therefore, in our research, we checked the association of *BDNF* gene expression with age in both groups of subjects, and we excluded age as a potential confounding factor.

COMT is an important modulator of different brain functions due to its role in dopamine metabolism. It is especially important in regulating dopamine neurotransmission in the prefrontal cortex [36], a part of the brain that plays a key role in cognitive functions and which is severely vulnerable to degeneration [79]. COMT is expressed not only in the CNS, but also in the periphery. The *COMT* gene is located on chromosome 22, and its expression is regulated by two promoters, P1 and P2 [80]. A longer mRNA which is transcribed from the P2 promoter encodes the membrane-bound COMT (MB-COMT), and a shorter mRNA originating from the P1 promoter encodes a shorter, soluble COMT (S-COMT). The MB-COMT is the dominant type of this enzyme in the brain tissue [81], and the S-COMT isoform is expressed widely in other tissues [80,82]. COMT affects cognitive control and functions in humans, and it is linked to the development of dementia [42]. Considering that the function of the dopamine system is disturbed in people diagnosed with AD, there is growing interest regarding the role of COMT in AD pathogenesis [83]. In the case of *COMT*, our results suggest no difference in the expression of this gene between AD patients and subjects diagnosed with MCI, and no effect of different clinical and demographic features, including age and gender, on the *COMT* mRNA expression. Thus far, the role of *COMT* in the pathogenesis of AD has mainly been focused on the association between its most commonly investigated functional polymorphism, Val158/108Met, and the development of AD and/or cognitive decline. Several studies have suggested the involvement of Val158/108Met with the risk of developing AD [84,85,86,87]. While some studies confirm that there is a connection [84,85,88], others have reported that there are no significant associations between this polymorphism and AD [40,86,89], including two meta-analyses [90,91]. These opposite findings could be explained by the complexity of AD and cognition, differences in diagnosis, numbers of patients, ethnical differences and different criteria for defining cognitive and behavioral symptoms [42]. Thus far, there has been no research on *COMT* gene expression in AD. Ni and colleagues compared the levels of *COMT* gene expression in healthy individuals and individuals diagnosed with schizophrenia, bipolar disorder and depression [92]. Their results indicate that, compared to healthy individuals, subjects with schizophrenia have a significantly lower relative expression of *COMT* mRNA, while its expression is increased in individuals with bipolar disorder [92]. The authors also suggested a significant positive correlation between *COMT* expression level and IQ scores, but only in healthy control subjects [92]. Our included subjects did not have schizophrenia, and since we did not control our subjects for the IQ scores, we are not able to confirm this correlation.

In order to further support the findings on gene expression, we investigated the DNA methylation levels of *BDNF* and *COMT* genes. Namely, DNA methylation is one of the plausible mechanisms that affects gene expression [93]. In the *BDNF* gene, the promotors I and IV, using nine different amplicons, were analyzed. We were not able to show prominent differences between DNA methylation levels in AD and MCI subjects. This might be explained by the fact that MCI can be considered as the pre-stage of AD, and the differences in methylation level could therefore already be too minuscule between the two groups or, alternatively, because we examined the whole amplicons instead of documenting individual CpGs, as has been performed in some studies. Namely, in a previous study on AD and controls, it was possible to detect significantly higher DNA methylation in AD patients compared to healthy subjects [94]. The CpGs investigated in this study are the CpGs in our amplicon BDNF_2 and they determined two CpGs that were significantly different between the two groups. Although in our case, we were not able to detect statistically significant differences, we did notice higher DNA methylation in AD patients compared to MCI subjects, as was observed in the study by Chang et al. [94]; furthermore, the methylation levels were also similar between the two studies. Another study investigated the promotor I and IV CGIs in controls and in MCI subjects with a 5-year follow up [95]. After 5 years, the MCI subjects were classified into two groups, an MCI-stable group and an AD-conversion group. The results showed higher DNA methylation of individual CpGs in the MCI group compared to the controls, and also in the AD-conversion group compared to the MCI-stable group [95]. In the study of Nagata et al. [96], the total DNA methylation of 20 CpGs in exon I showed higher methylation in the AD group compared to normal controls, while the individual DNA methylation differences showed a statically significant difference only in one CpG. They also found a positive correlation between one CpG and the duration of illness [96]. The methylation of *BDNF* and other putative AD genes were analyzed by Carboni et al., but no differences were determined between AD patients and controls for any of the studied genes [97]. Although all the listed studies were performed on blood samples, we still cannot point out a specific region that could be convincingly associated with AD and reproduced in independent studies. 

Similarly, as for *COMT* gene expression, there were no statistically significant results in our study regarding *COMT* methylation levels. We investigated the sequences around promotors 1 and 2, and also around the single-nucleotide polymorphism Val158/108Met, which, itself, is part of a CpG dinucleotide. The sequences around Val158/108Met and promotor 1 were found to be almost completely methylated (>95%), while the sequence around promotor 2 showed almost no methylation. Since promotor 1 regulates the expression of S-COMT, this was rather unexpected, also supporting the evidence that DNA methylation cannot always be regarded as an exclusive gene-expression-silencing mechanism [98]. We could not compare our results to any other study, as there are no published studies on AD and DNA methylation levels of the *COMT* gene so far.

Although we designed a study with two different clinical groups and were able to show changes in gene expression levels and DNA methylation, our study has some limitations that should be pointed out. In the study, we interrogated only specific sections in the DNA that have been previously shown to be sections associated with gene expression. However, the investigated genes are relatively large, and it would probably be even more informative if we looked at the complete sequence and determine the methylation level at all sites. Additionally, the number of transcripts of these genes is relatively high, and the interrogation of all transcripts would demand the use of more hydrolyzing probes or a transcriptomic approach. One of the limitations of the study is its cross-sectional design. A longitudinal follow-up would be useful since it would allow us to detect how many MCI subjects will develop dementia over time. Another limitation is the lack of a healthy control group without any cognitive decline. The strengths of the study are the inclusion of ethnically homogenous groups, and the adequate sample size and needed statistical power.

## 5. Conclusions and Future Directions

The results of this study confirm that the level of *BDNF* expression in the periphery is decreased in subjects with AD compared to individuals diagnosed with MCI. However, in the case of the *COMT* mRNA expression, the distinction between the two groups of subjects was not detected. Since the pathogenesis of AD begins many years before the onset of symptoms, the treatment is further complicated due to the cascade mechanisms that are activated as a result of the pathology of the disease. Earlier detection would facilitate therapy, and this is precisely why the goal of this research was to determine whether a reliable biomarker for the development of dementia can be established through the expression and methylation of *BDNF* and *COMT* in peripheral blood. The combined results from the gene expression analysis and DNA methylation analysis point to BDNF’s potential as a marker that could help distinguish between MCI and AD patients.

Although researchers and clinicians have produced an extensive amount of information on AD, from a morphological to a molecular level, we still lack knowledge on the underpinnings of its early development. In future, it would therefore be worth designing more studies on the early stages of cognitive impairment, such as subjective cognitive decline and mild cognitive impairment. Contemporary molecular approaches that allow for interrogation of the complete genome, epigenome and transcriptome could be applied and combined with powerful machine learning methods to discern key changes on a molecular level. These molecular players could be further tested in a clinical environment in order to identity the most specific and sensitive markers with potential for use in determination of the disease.

## Figures and Tables

**Figure 1 biomedicines-11-00235-f001:**
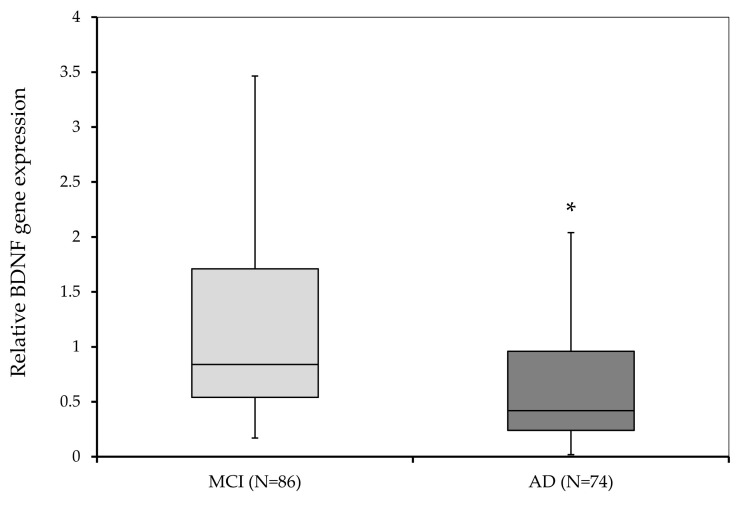
Relative *BDNF* gene expression in subjects with MCI and subjects diagnosed with AD. * *p* < 0.001.

**Figure 2 biomedicines-11-00235-f002:**
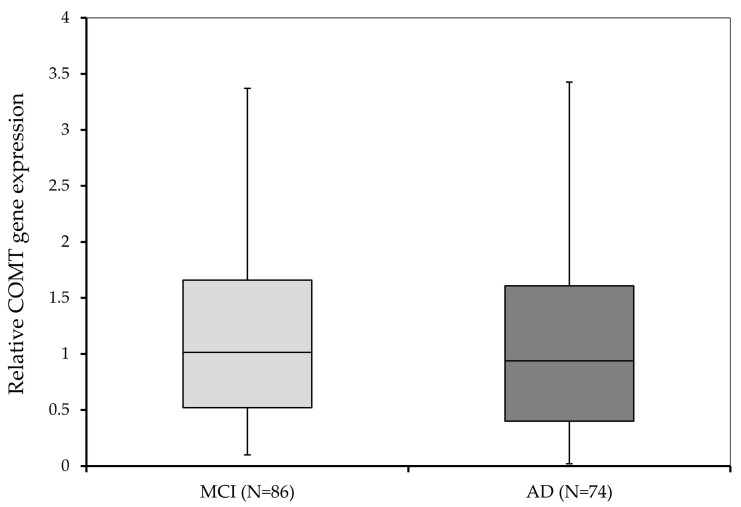
Relative *COMT* gene expression in subjects with MCI and subjects diagnosed with AD.

**Table 1 biomedicines-11-00235-t001:** The amplicon positions and lengths and the number of CpGs for the *BDNF* and *COMT* genes.

Amplicon	Position (Reference Genome Build hg19, Strand)	Target Amplicon Length (bp) *	Number of Interrogated CpGs
COMT_1	chr22:19951071-19951343	273	14
COMT_2	chr22:19929042-19929349	308	36
COMT_4	chr22:19950002-19950320	319	13
BDNF_1	chr11:27744260-27744605 (-)	346	22
BDNF_2	chr11:27743702-27743960 (-)	259	10
BDNF_3	chr11:27743454-27743762 (-)	309	20
BDNF_4	chr11:27741988-27742250 (-)	263	13
BDNF_5	chr11:27740916-27741131 (-)	216	16
BDNF_6	chr11:27740607-27740901 (-)	295	30
BDNF_7	chr11:27721638-27721854 (-)	217	19
BDNF_8	chr11:27722466-27722696 (-)	231	13
BDNF_9	chr11:27722209-27722487 (-)	279	23

* Target amplicon length without Illumina adapter overhang sequences at 5′ end of forward and reverse primers.

**Table 2 biomedicines-11-00235-t002:** Demographic and clinical data of subjects with MCI disorder and subjects with AD. All data are presented as median (range).

Characteristics	Participants		Mann–Whitney U Test	
	MCI	AD	U	*p*
Age (years)	71.0 (57.0–87.0)	79.0 (63.0–89.0)	5111.5	**<0.001**
BMI (kg/m^2^)	22.0 (18.4–32.4)	22.9 (18.5–31.9)	3355.5	0.550
Waist circumference (cm)	86.0 (71.0–101.0)	86.0 (72.0–99.0)	3173.0	0.975
Total cholesterol (mmol/L)	5.7 (3.2–8.8)	5.6 (3.2–8.8)	2936.5	0.400
HDL cholesterol (mmol/L)	1.3 (0.7–3.0)	1.3 (0.7–3.0)	3012.5	0.560
LDL cholesterol (mmol/L)	3.5 (0.8–5.6)	3.2 (0.8–5.8)	3048.5	0.647
Triglycerides	1.8 (0.7–6.7)	1.7 (0.7–6.7)	3048.0	0.645
Blood glucose (mmol/L)	5.5 (4.5–11.8)	5.6 (4.7–11.8)	3415.0	0.423
MMSE score	27.0 (21.0–28.0)	13.0 (10.0–24.0)	27.5	**<0.001**
CDT score	5.0 (1.0–5.0)	2.0 (1.0–5.0)	187.0	**<0.001**

AD—Alzheimer’s disease; CDT—Clock Drawing Test; HDL—high-density lipoproteins; BMI—body mass index; LDL—low-density lipoproteins; MCI—mild cognitive impairment; MMSE—Mini-Mental State Examination; N—number of participants.

**Table 3 biomedicines-11-00235-t003:** DNA methylation status of *BDNF* and *COMT* gene amplicons. DNA methylation values are presented as average DNA methylation percentage per studied group.

Gene	Amplicon Name	Number of DMCs/Number of Investigated CpGs	DNAm % ADs	DNAm % MCIs	DNAm % Difference	Mann–Whitney Test	
						U	*p*-Value
*BDNF*	BDNF_1	2/22	5.095	4.549	−0.546	196	0.5499
	BDNF_2	2/10	6.635	5.912	−0.723	36	0.3150
	BDNF_3	3/20	3.552	3.989	0.436	186	0.3963
	BDNF_4	0/13	11.39	11.36	−0.036	81	0.8798
	BDNF_5	1/16	7.019	6.803	−0.216	143	0.9729
	BDNF_6	0/30	6.654	6.957	0.302	473	0.9221
	BDNF_7	2/19	1.938	1.915	−0.023	167	0.7075
	BDNF_8	0/13	1.242	1.238	−0.004	75	0.6498
	BDNF_9	8/23	0.855	0.990	0.135	202	0.0778
*COMT*	COMT_1	0/14	96.00	96.02	0.017	111	0.9674
	COMT_2	0/36	0.1725	0.1730	0.001	617	0.7328
	COMT_4	2/13	95.99	95.78	−0.210	56	0.7969

DNAm—DNA methylation; ADs—subjects with Alzheimer’s disease; DMC—differentially methylated cytosine; MCIs—subjects with mild cognitive impairment.

## Data Availability

Not applicable.

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
