# Peer review of "Difference in Methylation and Expression of Brain-Derived Neurotrophic Factor in Alzheimer’s Disease and Mild Cognitive Impairment"

_biomedicines, 2023, doi:10.3390/biomedicines11020235_

Round 1
Reviewer 1 Report
12 December 2022
Regarding the review of manuscript ‘Difference in methylation and expression of BDNF and COMT in Alzheimer's disease and mild cognitive impairment’ by Kouter K et al., submitted to Biomedicines
Manuscript ID: biomedicines-2100243
Dear Authors,
Kouter and colleagues in the present research article entitled ‘Difference in methylation and expression of BDNF and COMT in Alzheimer's disease and mild cognitive impairment’, explored the level of brain derived neurotrophic factor (BDNF) expression in the periphery of subjects with Alzheimer’s disease (AD) compared to individuals diagnosed with mild cognitive impairment (MCI).
The main strength of this manuscript is that it addresses an interesting and timely question, providing a captivating interpretation and addressing the potential role of BDNF expression as a potential biomarker that could help distinguish between MCI and AD patients.
In general, I think the idea of this article is really interesting and the authors’ fascinating observations on this timely topic may be of interest to the readers of Biomedicines. However, some comments, as well as some crucial evidence that should be included to support the author’s argumentation, needed to be addressed to improve the quality of the manuscript, its adequacy, and its readability prior to the publication in the present form. My overall judgment is to publish this paper after the authors have carefully considered my suggestions below, in particular reshaping parts of the ‘Introduction’ and ‘Methods’ sections by adding more evidence.
Please consider the following comments:
1. Title: Please present the title self-explanatory and stating the most important parts of this study. Also, please avoid using abbreviations in the title.
2. Abstract: According to the Journal’s guidelines, the abstract should be a total of about 200 words maximum and should be presented as a single paragraph, without explicit subheadings. Please present the abstract, proportionally stating the background, the objectives, the methods, the results, and the conclusion. The background should include the general background (one to two sentences), the specific background (two to three sentences), and current issue addressed to this article (one sentence). The end of the result should include one to two sentences which put the result into a more general context. The conclusion should include one sentence describing the main message using such words like “Here we show”, the potential and the advance this article has provided in the field, and finally a broader perspective (two to three sentences) readily comprehensible to a scientist in any discipline.
3. A graphical abstract that will visually summarize the main findings of the manuscript is highly recommended.
4. Keywords: Please list ten keywords and use as many as possible in the title and the first two sentences of the abstract.
5. Introduction: The ‘Introduction’ section is well-written and nicely presented, with a good balance of descriptive text and information about pathologic substrates of AD. Nevertheless, I believe that more information on the pathophysiology and the core features of this disorder will provide a better and more accurate background, because as it stands, this information is not highlighted in the text. In this regard, I would suggest adding more information on pathological neural substrates of neurodegeneration in AD, specifically on structural as well as functional abnormalities of specific brain regions (i.e., hippocampus and prefrontal cortex), and on related and on related effects on patients’ cognitive impairments. In my opinion, authors could further explore significant structural brain alterations and impaired brain circuits in AD (https://doi.org/10.1038/s41380-021-01326-4; https://doi.org/10.17219/acem/146756), and focus on relationship between the molecular regulation of higher-order neural circuits and neuropathological alterations in this neurodegenerative disorder (https://doi.org/10.3390/cells11162607; https://doi.org/10.3390/biomedicines9050517).
6. Participants: Data about participants and information about clinical assessment for patients’ selection are not adequately explained. For this reason, I would ask the authors to specify inclusion criteria for patients involved in this study, like severity of disorder. Also, could the authors specify how did they estimate the exact number of participants and provide more information about the diagnostic tests used for clinical evaluation?
7. In my opinion, the 'Results’ section is well organized, but it seems to state statistical significance of findings in an excessively broad way. Thus, I believe that this section would benefit from a more detailed and precise rewriting, in order to ensure in-depth understanding of the findings.
8. I think the ‘Conclusions’ paragraph would benefit from some thoughtful as well as in-depth considerations by the authors, because as it stands, it lists down all the main findings of the research, without really stressing the theoretical significance of the study. The authors should make their efforts to explain the theoretical implication as well as the translational application of their research.
9. In according to the previous comment, I would ask the authors to include a proper and defined ‘Limitations and future directions’ section before the end of the manuscript, in which authors can describe in detail and report all the technical issues brought to the surface.
10. Tables: According to the Journal’s guidelines, please provide an explanatory caption before each table within the text.
11. Figures: Please present the figures in color.
12. References: Please correct the references according to the journal’s guidelines (https://www.mdpi.com/journal/biomedicines/instructions) and place the doi numbers.
Overall, the manuscript contains 3 tables, 2 figures and 90 references. In my opinion, the manuscript might carry important value describing addressing the potential role of BDNF expression as a potential biomarker that could help distinguish between MCI and AD patients.
I hope that, after these careful revisions, this paper can meet the Journal’s high standards for publication.
I am available for a new round of revision of this article.
I declare no conflict of interest regarding this manuscript.
Best regards,
Reviewer
Author Response
Reviewer 1
Dear Authors,
Kouter and colleagues in the present research article entitled ‘Difference in methylation and expression of BDNF and COMT in Alzheimer's disease and mild cognitive impairment’, explored the level of brain derived neurotrophic factor (BDNF) expression in the periphery of subjects with Alzheimer’s disease (AD) compared to individuals diagnosed with mild cognitive impairment (MCI).
The main strength of this manuscript is that it addresses an interesting and timely question, providing a captivating interpretation and addressing the potential role of BDNF expression as a potential biomarker that could help distinguish between MCI and AD patients.
In general, I think the idea of this article is really interesting and the authors’ fascinating observations on this timely topic may be of interest to the readers of Biomedicines. However, some comments, as well as some crucial evidence that should be included to support the author’s argumentation, needed to be addressed to improve the quality of the manuscript, its adequacy, and its readability prior to the publication in the present form. My overall judgment is to publish this paper after the authors have carefully considered my suggestions below, in particular reshaping parts of the ‘Introduction’ and ‘Methods’ sections by adding more evidence.
Please consider the following comments:
- Title: Please present the title self-explanatory and stating the most important parts of this study. Also, please avoid using abbreviations in the title.
Thank you for your comment. We have altered the title in order to point out just the main and significant findings of the study. Now the title states: “Difference in methylation and expression of brain-derived neurotrophic factor in Alzheimer's disease and mild cognitive impairment”.
- Abstract: According to the Journal’s guidelines, the abstract should be a total of about 200 words maximum and should be presented as a single paragraph, without explicit subheadings. Please present the abstract, proportionally stating the background, the objectives, the methods, the results, and the conclusion. The background should include the general background (one to two sentences), the specific background (two to three sentences), and current issue addressed to this article (one sentence). The end of the result should include one to two sentences which put the result into a more general context. The conclusion should include one sentence describing the main message using such words like “Here we show”, the potential and the advance this article has provided in the field, and finally a broader perspective (two to three sentences) readily comprehensible to a scientist in any discipline.
Thank you for your comment. We have shortened the abstract and now it has a total of 196 word.
- A graphical abstract that will visually summarize the main findings of the manuscript is highly recommended.
We have added the graphical abstract.
- Keywords: Please list ten keywords and use as many as possible in the title and the first two sentences of the abstract.
The list of keywords has been expanded and has been used in the title and abstract.
- Introduction: The ‘Introduction’ section is well-written and nicely presented, with a good balance of descriptive text and information about pathologic substrates of AD. Nevertheless, I believe that more information on the pathophysiology and the core features of this disorder will provide a better and more accurate background, because as it stands, this information is not highlighted in the text. In this regard, I would suggest adding more information on pathological neural substrates of neurodegeneration in AD, specifically on structural as well as functional abnormalities of specific brain regions (i.e., hippocampus and prefrontal cortex), and on related and on related effects on patients’ cognitive impairments. In my opinion, authors could further explore significant structural brain alterations and impaired brain circuits in AD (https://doi.org/10.1038/s41380-021-01326-4; https://doi.org/10.17219/acem/146756), and focus on relationship between the molecular regulation of higher-order neural circuits and neuropathological alterations in this neurodegenerative disorder (https://doi.org/10.3390/cells11162607; https://doi.org/10.3390/biomedicines9050517).
The introduction has been expanded as requested.
- Participants: Data about participants and information about clinical assessment for patients’ selection are not adequately explained. For this reason, I would ask the authors to specify inclusion criteria for patients involved in this study, like severity of disorder. Also, could the authors specify how did they estimate the exact number of participants and provide more information about the diagnostic tests used for clinical evaluation?
We have added the additional inclusion criteria: “Inclusion criteria were in- and out-patients who signed informed written consent. The subjects involved in the research had not previously taken any antidementia drugs and were not related to each other.” Severity of the disorder was not one of the inclusion criteria.
As stated in the Statistical analysis: “G∗Power 3 Software [41] was used to calculate the needed sample size and statistical power. With expected effect size = 0.50, and statistical power set to 0.85, the required sam-ple size was N=154 for Mann-Whitney U test and N=150 for correlation. Our study in-cluded a total of 160 subjects which is above the sample size needed to detect differences between groups.”
- In my opinion, the 'Results’ section is well organized, but it seems to state statistical significance of findings in an excessively broad way. Thus, I believe that this section would benefit from a more detailed and precise rewriting, in order to ensure in-depth understanding of the findings.
Thank you for your comment. We tried to point out the main results and we think that the section is really easy to follow. Main findings are also summarized once again at the beginning of the Discussion. We would appreciate if the reviewer could point out more precisely what should be rewritten in more details and more precisely. We have tried to organize the Results in order to give the main differences in demographic and clinical data for the participants. Then we gave separately the results regarding BDNF and COMT expression. Bot subsections also include the results regarding the possible influence of demographic and clinical parameters on the gene expression levels, with more details given in supplementary materials. And the last subsection refers to methylation analysis with the main results presented in the Table 3.
- I think the ‘Conclusions’ paragraph would benefit from some thoughtful as well as in-depth considerations by the authors, because as it stands, it lists down all the main findings of the research, without really stressing the theoretical significance of the study. The authors should make their efforts to explain the theoretical implication as well as the translational application of their research.
Thank you for your comment. We have extended the conclusion and added future directions.
- In according to the previous comment, I would ask the authors to include a proper and defined ‘Limitations and future directions’ section before the end of the manuscript, in which authors can describe in detail and report all the technical issues brought to the surface.
As proposed we added limitations in the end of the Discussion and we wrote Future directions together with conclusion.
- Tables: According to the Journal’s guidelines, please provide an explanatory caption before each table within the text.
Thank you for your comment. We have added explanatory caption for all tables in the manuscript.
- Figures: Please present the figures in color.
Thank you for your comment. Considering that there is no need for the figures to be in color, that is, it would not increase the informativeness of the figures themselves, and the journal itself prefers black and white graphics, when possible, we decided to keep the pictures in black and white. If the reviewer insists, and the journal agrees, we will transform the figures and present them in color.
- References: Please correct the references according to the journal’s guidelines (https://www.mdpi.com/journal/biomedicines/instructions) and place the doi numbers.
We have added DOI number where available.
Reviewer 2 Report
An interesting article. I have 2 minor suggestions:
1) In page 9 (results) the authors comment that "eight out of 23 investigated CpG sites showed significant difference in DNA methylation levels". However, according to the tables and the discussion, there are a non-significant trend towars statistical significance (p=0.0778). Please, amend.
2) A brief paragraph on the strengths and limitations of the study seems appropriate
Author Response
An interesting article. I have 2 minor suggestions:
1) In page 9 (results) the authors comment that "eight out of 23 investigated CpG sites showed significant difference in DNA methylation levels". However, according to the tables and the discussion, there are a non-significant trend towars statistical significance (p=0.0778). Please, amend.
Thank you for this comment. We further emphasized the difference between the values of individual CpG sites and the values of the amplicon as a whole. The paragraph has been improved:
»Analyzing mean DNA methylation levels per amplicon, no statistically significant changes were observed between the average DNA methylation levels between subjects with AD and subject with MCI. When comparing the DNA methylation status of CpGs between subject with AD or MCI, we did detect multiple single base pair DMCs. We detected the highest number of DMCs in the BDNF gene amplicon BDNF_9, where eight out of the 23 investigated CpG sites showed statistically significant difference in DNA methylation levels. However, when looking at the mean DNA methylation level, the level of BDNF_9 per fold only showed a trend towards statistical significance. The differences in the DNA methylation levels between subjects with AD and subjects with MCI were modest (under 1%), which opens the question of their biological relevance.«
2) A brief paragraph on the strengths and limitations of the study seems appropriate
As proposed, we have added a paragraph on strengths and limitations in the end of the Discussion: “Although we designed a study with two different clinical groups and were able to show changes in gene expression levels and DNA methylation, our study has some limitations that should be pointed out. In the study we interrogated only specific sections in the DNA that have been previously shown as sections associated with gene expression. However, the investigated genes are relatively large, and probably it would be even more informative, if we looked at the complete sequence and determine the methylation level at all sites. Also, the number of transcripts of these genes is relatively high, and interrogation of all transcripts would demand use of more hydrolyzing probes or a transcriptomic approach. One of the limitations of the study is its cross-sectional design. The longitudinal follow up would be useful since it would allow us to detect how many MCI subjects will develop dementia in time. Another limitation is the lack of healthy control group without any cognitive decline. Strengths of the study are the inclusion of ethnically homogenous groups, adequate sample size and needed statistical power.”
Round 2
Reviewer 1 Report
21 December 2022
Regarding the 2nd review of manuscript ‘Difference in methylation and expression of BDNF and COMT in Alzheimer's disease and mild cognitive impairment’ by Kouter K et al., submitted to Biomedicines
Manuscript ID: biomedicines-2100243
Dear Authors,
I am pleased to see that the authors took my comments seriously and solved many issues I raised in the previous round of the pee-review session. Currently, the manuscript is a well written and nicely presented research paper studying the level of brain derived neurotrophic factor expression in the periphery of subjects with Alzheimer’s disease compared to individuals diagnosed with mild cognitive impairment. That said, I just leave a couple of comments which I believe useful to improve the quality of this manuscript to finalize my part of the review session.
Comments:
1. Introduction: This section has substantially improved and is well written; nevertheless, more information on pathological neural substrates of neurodegeneration and reduced resilience of neuroplasticity in neurodegenerative diseases may help ensure the reliability and the integrity of evidence that the authors present the main constructs of this topic and put forward the argument and the question they have tried to reveal (https://doi.org/10.3389/fnbeh.2022.998714; https://doi.org/10.17219/acem/149897; https://doi.org/10.3390/ijms23136991; doi: 10.3390/ijms222413384.).
2. References: Some of journal names remain expanded and not abbreviated without period (.). Please correct them.
Overall, the manuscript contains 3 tables, 2 figures and 95 references. The manuscript carries important value describing showing the potential role of brain-derived neurotrophic factor expression as a potential biomarker that could help distinguish between mild cognitive impairment and Alzheimer’s disease.
After careful revisions, this paper meets the Journal’s high standards for publication.
I declare no conflict of interest regarding this manuscript.
Best regards,
Reviewer
Author Response
Reviewer 2, 2nd review
Manuscript ID: biomedicines-2100243
Dear Authors,
I am pleased to see that the authors took my comments seriously and solved many issues I raised in the previous round of the pee-review session. Currently, the manuscript is a well written and nicely presented research paper studying the level of brain derived neurotrophic factor expression in the periphery of subjects with Alzheimer’s disease compared to individuals diagnosed with mild cognitive impairment. That said, I just leave a couple of comments which I believe useful to improve the quality of this manuscript to finalize my part of the review session.
Comments:
- Introduction: This section has substantially improved and is well written; nevertheless, more information on pathological neural substrates of neurodegeneration and reduced resilience of neuroplasticity in neurodegenerative diseases may help ensure the reliability and the integrity of evidence that the authors present the main constructs of this topic and put forward the argument and the question they have tried to reveal (https://doi.org/10.3389/fnbeh.2022.998714; https://doi.org/10.17219/acem/149897; https://doi.org/10.3390/ijms23136991; doi: 10.3390/ijms222413384.).
Thank you for all suggestions. We have improved the text as proposed and we have colored the new text in turquoise.
- References: Some of journal names remain expanded and not abbreviated without period (.). Please correct them.
We checked the references once more and corrected them. According to Instructions for authors saying ‘If you are not sure how to abbreviate a particular journal title, please leave the entire title. The Editorial Office will abbreviate those journal titles appropriately.’, we left some journal names in full.